# Neighbor predation linked to natural competence fosters the transfer of large genomic regions in *Vibrio cholerae*

Noémie Matthey[1], Sandrine Stutzmann[1], Candice Stoudmann[1], Nicolas Guex[2†], Christian Iseli[2†], Melanie Blokesch[1*]

[1]Laboratory of Molecular Microbiology, Global Health Institute, School of Life Sciences, Ecole Polytechnique Fédérale de Lausanne (Swiss Federal Institute of Technology Lausanne; EPFL), Lausanne, Switzerland; [2]Swiss Institute of Bioinformatics, Lausanne, Switzerland

**Abstract** Natural competence for transformation is a primary mode of horizontal gene transfer. Competent bacteria are able to absorb free DNA from their surroundings and exchange this DNA against pieces of their own genome when sufficiently homologous. However, the prevalence of non-degraded DNA with sufficient coding capacity is not well understood. In this context, we previously showed that naturally competent *Vibrio cholerae* use their type VI secretion system (T6SS) to actively acquire DNA from non-kin neighbors. Here, we explored the conditions of the DNA released through T6SS-mediated killing versus passive cell lysis and the extent of the transfers that occur due to these conditions. We show that competent *V. cholerae* acquire DNA fragments with a length exceeding 150 kbp in a T6SS-dependent manner. Collectively, our data support the notion that the environmental lifestyle of *V. cholerae* fosters the exchange of genetic material with sufficient coding capacity to significantly accelerate bacterial evolution.
DOI: https://doi.org/10.7554/eLife.48212.001

*For correspondence:
melanie.blokesch@epfl.ch

Present address: †Bioinformatics Competence Center, University of Lausanne, Lausanne, Switzerland

Competing interests: The authors declare that no competing interests exist.

## Introduction

The causative agent of the diarrheal disease cholera, *Vibrio cholerae*, is responsible for seven major pandemics since 1817, one of which is still ongoing. Due to its ability to rapidly spread in contaminated water, cholera poses a serious world health risk, affecting between 1 and 4 million people and causing 21,000–143,000 deaths per year, especially in poor or underdeveloped countries (*WHO, 2019*). Many disease-causing bacteria have developed mechanisms for rapidly evolving in response to environmental pressures, and these rapid changes are often responsible for the formation of new serogroups with pandemic potential. One way in which *V. cholerae* acquire new phenotypes is through horizontal gene transfer (HGT), which is the direct movement of DNA from one organism to another. A major mode of HGT is natural competence for transformation in which bacteria are able to absorb free DNA from their surroundings using their competence-induced DNA-uptake complex (*Chen and Dubnau, 2004*; *Johnston et al., 2014*; *Matthey and Blokesch, 2016*). When sufficient homology is present between the incoming DNA and the bacterial genome, the absorbed genetic material can be integrated into the genome via double homologous recombination at the expense of the initial DNA region. As an example of the significant power of this natural competence for gene uptake, we previously witnessed the gain of an ~40 kbp O139-antigen cluster at the expense of the original ~30 kbp O1-antigen cluster through natural transformation (followed by strong selective pressure exerted by antibiotics or phages; *Blokesch and Schoolnik, 2007*), which significantly changed the phenotypes of these bacteria. And while Griffith's experiment in 1928 unambiguously proved that transformation contributes to evolution and pathogen emergence, the

general prevalence of non-degraded DNA with sufficient coding capacity has been questioned (*Nielsen et al., 2007*; *Overballe-Petersen et al., 2013*; *Croucher et al., 2016*), drawing inquiries as to whether this mode of HGT could be responsible for the major changes causing pandemic strains to emerge.

The induction of competence in *V. cholerae* is tightly regulated (recently reviewed by *Metzger and Blokesch, 2016*). Briefly, upon growth on the (molted) chitin-rich exoskeletons of zooplankton (*Pruzzo et al., 2008*), the most abundant polysaccharide in the aquatic environment and therefore an important carbon source for chitinolytic bacteria (*Gooday, 1990*), the expression pattern of *V. cholerae* is altered (*Meibom et al., 2004*) to render it naturally competent for genetic transformation (*Meibom et al., 2005*). Initially, when chitin degradation products are sensed by *V. cholerae*, it produces the regulatory protein TfoX (*Li and Roseman, 2004*; *Meibom et al., 2004*; *Meibom et al., 2005*; *Yamamoto et al., 2011*; *Dalia et al., 2014*; *Yamamoto et al., 2014*). This competence activator positively regulates the expression of the major DNA-uptake machinery in the cell (*Matthey and Blokesch, 2016*), providing a direct connection between growth on chitin and competence activation. Apart from TfoX, natural competence and transformation also depend on the master regulator of quorum sensing, HapR, in two ways: i) HapR acts as repressor of *dns*, which encodes an extracellular nuclease that inhibits transformation (*Blokesch and Schoolnik, 2008*); and ii) HapR together with TfoX co-activates the transcription factor QstR, which further represses *dns* as well as activates several DNA-uptake genes (*Lo Scrudato and Blokesch, 2013*; *Jaskólska et al., 2018*).

While the chitin-induced DNA-uptake complex of *V. cholerae* is able to absorb DNA from the surrounding (*Seitz and Blokesch, 2013*; *Seitz and Blokesch, 2014*; *Seitz et al., 2014*; *Ellison et al., 2018*; *Adams et al., 2019*), environmental DNA is often heavily degraded and therefore short in size (*Nielsen et al., 2007*; *Overballe-Petersen et al., 2013*). In addition, free DNA is thought to originate from dead and therefore less fit bacteria, which renders the coding part of such genetic material non-favorable for naturally competent bacteria (*Redfield, 1988*). In line with these arguments, we recently showed that *V. cholerae* does not solely rely on randomly released environmental DNA. Instead, it actively acquires 'fresh' DNA from healthy, living bacteria through kin-discriminatory neighbor predation (*Borgeaud et al., 2015*), which, conceptually, also occurs in other naturally competent bacteria (*Veening and Blokesch, 2017*). Neighbor predation in *V. cholerae* is accomplished by a contractile injection system known as the type VI secretion system (T6SS) that transports toxic effector proteins into prey (*Ho et al., 2014*; *Cianfanelli et al., 2016*; *Galán and Waksman, 2018*; *Taylor et al., 2018*). Intriguingly, the T6SS of pandemic *V. cholerae* is exquisitely co-regulated with its DNA-uptake machinery in a TfoX-, HapR-, and QstR-dependent manner when the bacterium grows on chitin (*Borgeaud et al., 2015*; *Watve et al., 2015*; *Metzger et al., 2016*; *Jaskólska et al., 2018*), which increases the chances of the competent bacterium to take up freshly released DNA compared to free-floating 'unfit' DNA. Notably, this coupling of competence and type VI secretion is also conserved in several non-cholera vibrios (*Metzger et al., 2019*).

In the current study, we determined the extent of the absorbed and chromosomally-integrated prey-derived DNA. Previous studies had scored transformation events in other naturally competent Gram-negative bacteria such as *Haemophilus influenzae*, *Helicobacter pylori*, and *Neisseria meningitidis* (*Mell et al., 2014*; *Bubendorfer et al., 2016*; *Alfsnes et al., 2018*). These former studies, however, relied on the supplementation of large quantities of purified DNA (with up to 50 donor genome equivalents per cell; *Bubendorfer et al., 2016*) at the peak of the organism's competence program (*Mell et al., 2014*). Such an approach neither recapitulates the natural onset of competence nor discloses the fate of the DNA that is released from dying cells. Thus, to address these points and to mimic natural settings, we determined the frequency and extent of DNA exchanges under chitin-dependent co-culture conditions of two non-clonal *V. cholerae* strains. We show that the DNA transfer frequency is significantly enhanced in T6SS-positive compared to T6SS-negative strains and that large genomic regions are transferred from the killed prey to the competent acceptor bacterium.

## Results and discussion

### The T6SS fosters horizontal co-transfer events encompassing two selective markers

To compare the absorption of T6SS-mediated prey-derived DNA as opposed to environmental DNA (released through, for example, random lysis), we first scored the transformability of T6SS-positive (wild-type [WT] predator) and T6SS-negative (acceptor) *V. cholerae* strains, which would allow us to directly measure the contribution of the T6SS on gene uptake. These two strains were co-cultured with non-kin prey (donor) bacteria that were all derived from the environmental isolate Sa5Y (*Keymer et al., 2007*; *Miller et al., 2007*; *Borgeaud et al., 2015*; *Matthey et al., 2018*) and contained two antibiotic resistance genes in their genomes: 1) An *aph* cassette (Kan$^R$), which was integrated in the *vipA* gene on the small chromosome (chr 2); and 2) a *cat* cassette (Cm$^R$), which was inserted at variable distances from the *aph* cassette on the same chromosome or, alternatively, on the large chromosome (chr 1). As shown in *Figure 1*, the WT predator strain efficiently absorbed and integrated the prey-released resistance cassettes (*aph* or *cat*), while the transformation efficiency for the T6SS-defective acceptor strain was significantly reduced (by 97.8% and 99.2% for *aph* and *cat*, respectively) (*Figure 1A*). Moreover, comparable frequencies were observed for both selective markers, suggesting that their acquisition does not significantly affect the strains' fitness under non-selective conditions. We tested whether these transfer events were indeed competence-mediated and not based on other modes of HGT using a strain with a competence-related DNA import deficiency in that it lacked the competence protein ComEA that reels external DNA into the periplasm (*Seitz et al., 2014*). This *comEA*-minus strain was never transformed under these predator-prey co-culture conditions, confirming that the gene transfer did depend entirely on natural competence.

Next, we scored the frequencies of transformants that had adopted resistance against both antibiotics, which would show the possibility of two transformation events or the transfer of a large piece of DNA (indicated by the distance between the two genes on the same chromosome). These transformations occurred, as expected, at lower rates compared to single-resistant clones and were mostly below the limit of detection for the T6SS-minus acceptor strain (*Figure 1B*). Interestingly, we observed a gradual decrease in the frequencies the further the two resistance genes were apart from each other on the same chromosome, while a sharp drop occurred in the number of recovered transformants when the two resistance genes were carried on the two separate prey chromosomes (*Figure 1B*). While the latter scenario unambiguously requires at least two separate DNA-uptake events, the former, in which the resistance markers are carried in cis, could reflect a mix between single and multiple DNA absorption and integration events. When purified genomic DNA was instead provided as the transforming material to simplify the experiment and provide measurable results for all conditions, the in cis double-resistance acquisition efficiencies reached a comparable range to the in trans efficiencies when the two resistance genes were separated by at least 100 kbp. This suggested that the more efficient transformations of less than 100 kbp likely often occurred through a single acquisition (*Figure 1C*). Furthermore, the WT predator and T6SS-minus acceptor behaved similarly when purified DNA was provided, which makes sense as the need for active DNA release through neighbor predation was eliminated. Based on these data and the fact that the double-acquisition rates for the T6SS-minus acceptor strain were mostly below the detection limit in the prey scenario (*Figure 1B*), we hypothesized that neighbor predation might foster the transfer of long DNA stretches, which frequently exceeded 50 kbp and therefore carry significant coding capacity. Moreover, the significantly lower double-acquisition rates when both cassettes are located in trans (e.g., on two different chromosomes) for the co-culturing conditions compared to the supplementation of purified gDNA (*Figure 1—figure supplement 1*) led us speculate that long DNA stretches might be released from the killed prey thereby saturating the system against uptake of additional fragments.

### Comparative genomics of pandemic strain A1552 and environmental isolate Sa5Y

To test our hypothesis that the T6SS contributes to the horizontal transfer of large DNA fragments, we used a whole-genome sequencing (WGS) approach to properly outline the transferred DNA regions. To do this using WGS, we first needed to characterize the genomes of both the predator/acceptor (A1552) and the prey/donor (Sa5Y) strains for which long-read PacBio sequencing data and

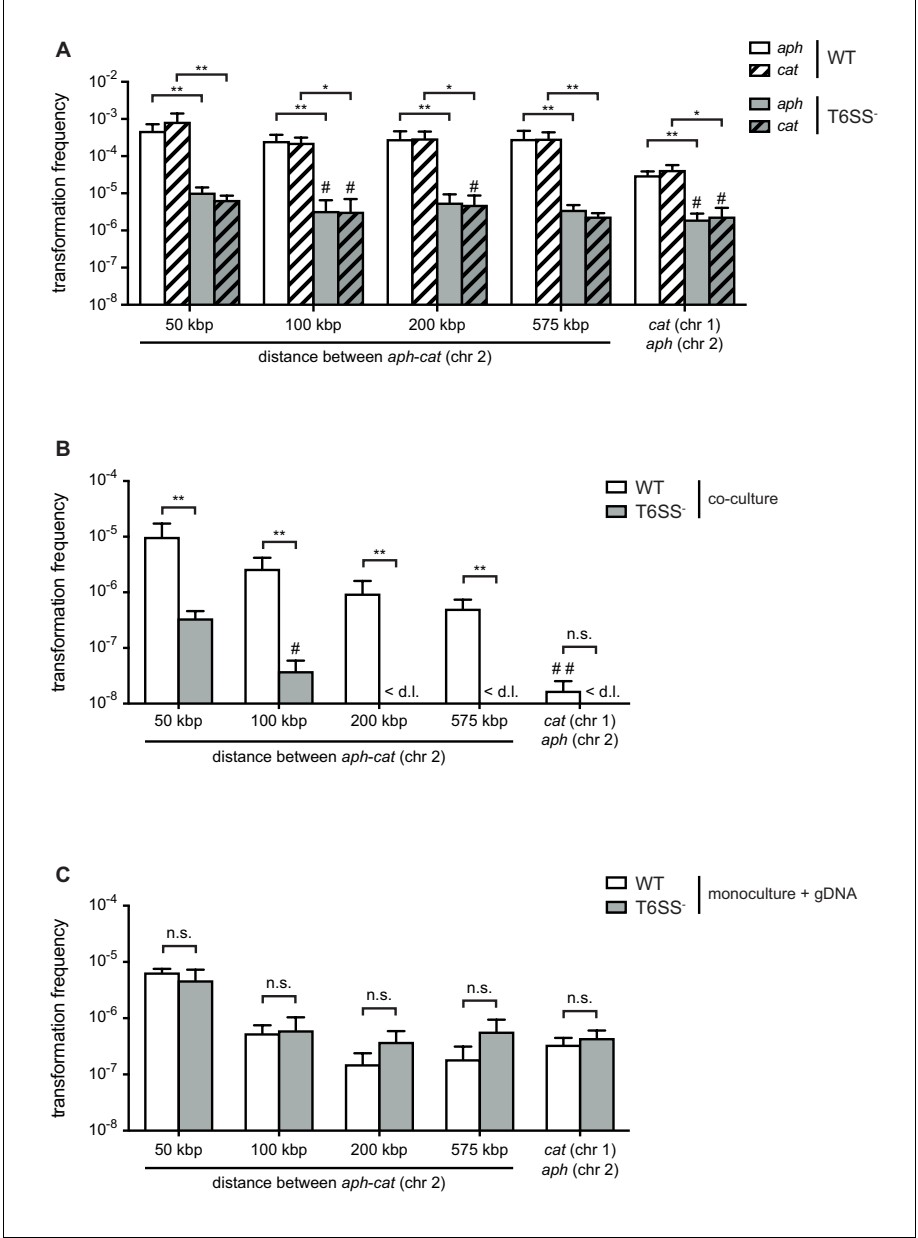

**Figure 1.** Type VI secretion system (T6SS) enhances horizontal gene transfer (HGT) of single- and double-resistance cassettes if carried in cis. (**A–B**) Transformation occurs in predator/prey co-cultures. To induce natural competence, the WT or a T6SS-negative derivative (A1552ΔvasK; T6SS ⁻) was co-cultured on chitin with different prey strains (Sa5Y-derived) that carried two antibiotic resistance cassettes: *aph* in *vipA* (chr 2) and *cat* at variable distances from *aph* on the same chromosome or on chr 1, as indicated on the X-axis. Transformation frequencies (*Y*-axis) indicate the number of transformants that acquired (**A**) a single resistance cassette or (**B**) both resistance cassettes divided by the total number of predator colony forming units (CFUs). (**C**) Natural transformation is not impaired in the T6SS⁻ acceptor strain. Purified genomic DNA (gDNA) was added to competent WT or T6SS⁻ strains. (**A–C**) Data represent the average of three independent biological experiments (± SD, as depicted by the error bars). For values in which one (#) or two (##) experiments resulted in the absence of transformants, the detection limit was used to calculate the average. <d .l., below detection limit. Statistical significance is indicated (*p<0.05; **p<0.01; n.s., not significant).

DOI: https://doi.org/10.7554/eLife.48212.002

The following source data and figure supplement are available for figure 1:

**Source data 1.** Raw data for *Figure 1*.

DOI: https://doi.org/10.7554/eLife.48212.004

*Figure 1 continued on next page*

*Figure 1 continued*

**Figure supplement 1.** Combined data from *Figure 1B and C* comparing double-resistance acquisition efficiencies in co-cultures or in gDNA-supplemented monocultures.

DOI: https://doi.org/10.7554/eLife.48212.003

de novo assemblies without further analysis were recently announced (*Matthey et al., 2018*)(*Figure 2*). A1552 is a pandemic O1 El Tor strain (*Yildiz and Schoolnik, 1998*) belonging to the LAT-1 sublineage of the West-African South American (WASA) lineage of seventh pandemic *V. cholerae* strains (*Domman et al., 2017*) while strain Sa5Y was isolated from the Californian Coast (*Keymer et al., 2007*; *Miller et al., 2007*). To understand their genomic arrangements, we also compared these strains to the reference sequence of *V. cholerae* (O1 El Tor strain N16961; *Heidelberg et al., 2000*) and a re-sequenced laboratory stock of the latter. Details on the comparative genomics between the three pandemic strains (N16961 [*Heidelberg et al., 2000*], the newly sequenced and de novo-assembled genome sequence of the laboratory stock of N16961, and

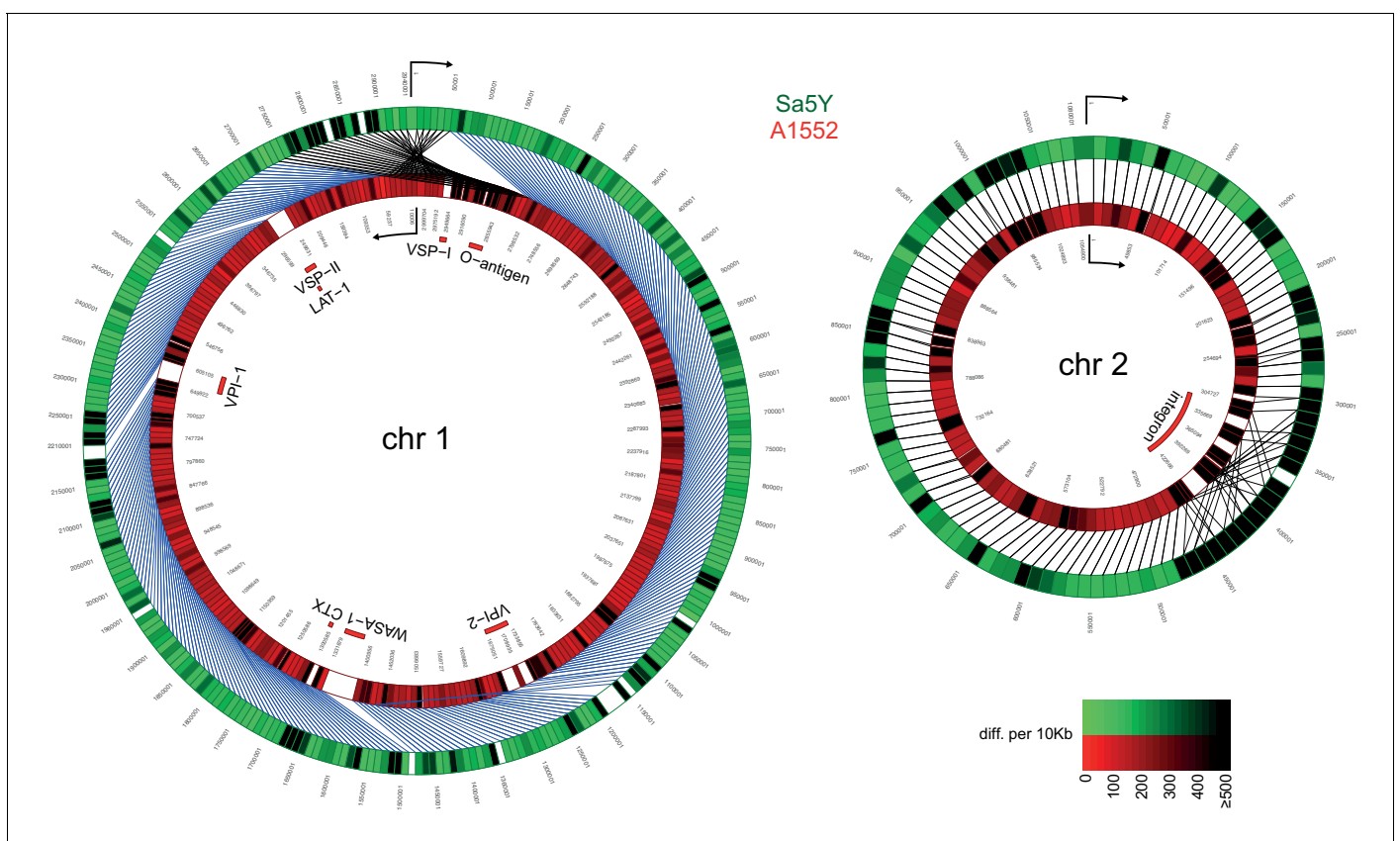

**Figure 2.** Comparative genomics of pandemic strain A1552 and the environmental isolate Sa5Y. The genomic sequences of chr 1 and 2 of Sa5Y (green) were segmented in 10-kbp-long fragments and aligned against the respective chromosome of the reference A1552 (red). To simplify visualization, chr 1 of strain A1552 was inverted and plotted counter-clockwise relative to Sa5Y (due to the large inversion in this strain; see Materials and methods section), as indicated by the arrow. To represent the differences between the two genomes, a color intensity scale was used that corresponded to the number of differences (SNP or indel), from 0 to ≥500 as measured per 10 kbp fragment (same values are indicated for both colors). White regions show no homology. Important genomic features of pandemic *V. cholerae* are highlighted inside the rings.

DOI: https://doi.org/10.7554/eLife.48212.005

The following figure supplements are available for figure 2:

**Figure supplement 1.** Comparative genomics of *V. cholerae* reference strain N16961 and a newly sequenced laboratory stock of the same strain.

DOI: https://doi.org/10.7554/eLife.48212.006

**Figure supplement 2.** Comparative genomics of pandemic *V. cholerae* strains N16961 and A1552.

DOI: https://doi.org/10.7554/eLife.48212.007

A1552) are provided in the Material and methods section and as *Figure 2—figure supplements 1* and *2*. We expected to see significant differences in the pandemic A1552 strain compared to the environmental isolate Sa5Y in terms of the absence/presence of genomic features and single nucleotide polymorphisms (SNPs) in core genes that would allow us to measure HGT events occurring between the strains, and several of these major differences are highlighted here. Indeed, as expected from its non-clinical origin, the environmental isolate lacked several genomic regions, including those that encode major virulence features, namely *Vibrio* pathogenicity islands 1 and 2 (VPI-1, VPI-2), *Vibrio* seventh pandemic islands I and II (VSP-I, VSP-II; *Dziejman et al., 2005*), the cholera toxin prophage CTX (*Waldor and Mekalanos, 1996*), and the WASA-1 element. In addition, the strain's O-antigen cluster differed significantly from the O1-encoding genes of pandemic strain A1552 (*Figure 2*). The region that differed the most between both strains was the integron island, which is consistent with the role of this assembly platform in fostering the incorporation of exogenous open reading frames (*Mazel, 2006*). Given these major differences between strain A1552 and Sa5Y and, in addition, an overall SNP frequency of approximately 1 in 55 nucleotides for conserved genes, we concluded that HGT events occurring between these two strains on chitinous surfaces could be precisely scored using short-read sequencing. Apart from this important genomic information, we also noted that the pandemic strains as well as Sa5Y contained previously unrecognized rRNA operons, with nine or ten rRNA clusters in total compared to the initially reported eight (*Heidelberg et al., 2000*).

## Released DNA from T6SS-killed prey leads to the transfer of large genomic regions

As our previous study witnessed gene transfers between *V. cholerae* bacteria (*Blokesch and Schoolnik, 2007*) though neither scored the full extent of the transferred DNA region nor took T6SS-mediated neighbor predation into consideration, we sought to next determine how much genetic material would be absorbed and integrated by competent *V. cholerae* upon neighbor predation. To do this, we co-cultured the predator (A1552) and prey (Sa5Y) strains on chitinous surfaces for 30 hr without any deliberate selection pressure. To be able to afterwards screen for the transfer of at least one gene, we first integrated an *aph* cassette within the *vipA* of strain Sa5Y, which concomitantly deactivated the prey's T6SS, to select kanamycin-resistant transformants of strain A1552. Using this system, resistant transformants of A1552 were selected at an average frequency of $1.8 \times 10^{-4}$ after the 30 hr co-culturing on chitin, and 20 of those transformants were randomly picked for further analysis. After three independent experiments, the whole genome of each of the 60 transformants was sequenced, and the reads were mapped to either the predator's or the prey's genome sequence (see Materials and methods section for detailed bioinformatic analysis). As shown in *Figure 3*, apart from the common acquisition of the *aph* resistance cassette, the location and the size of the prey-donated genomic region differed significantly between most transformants. Previous estimates of the average length of total acquired DNA were made in experiments using purified donor gDNA and were considered to be ~23 kbp (*Miller et al., 2007*). Importantly, we observed in these new experiments that the average length of the total acquired DNA, meaning the DNA surrounding the *aph* cassette plus any transferred regions elsewhere on either of the two chromosomes, was almost 70 kbp and therefore significantly larger than the previous estimates. Around 15% of all transformants acquired and integrated more than 100 kbp (*Figure 3B*), which was previously considered unlikely due to rareness of such long DNA fragments in the environment. Consistent with the principle of natural transformation, it should be noted that the new DNA was acquired through double homologous recombination such that it replaced the initial DNA region and the overall genome size did not significantly change. Further analysis indicated that about 50% of the strains experienced a single HGT event around the *aph* cassette, while the others exchanged regions in up to eight different locations on the two chromosomes (*Figure 3C*). Finally, we analyzed the length of continuous DNA stretches that were acquired from the prey and observed that those ranged from a few kbp up to 168 kbp (*Figure 3D*). Collectively, these data indicate that *V. cholerae* can acquire large genomic regions from killed neighbors with an average exchange of more than 50 kbp or ~50 genes. Notably, our observed recombination events were significantly larger than the mean recombination event size of ~4 kbp that was recently reported for surface-grown *Streptococcus pneumoniae* (*Cowley et al., 2018*). In their study, Cowley et al. observed that competence-mediated homologous recombination was more likely to occur upon direct cell-to-cell contact, which the authors

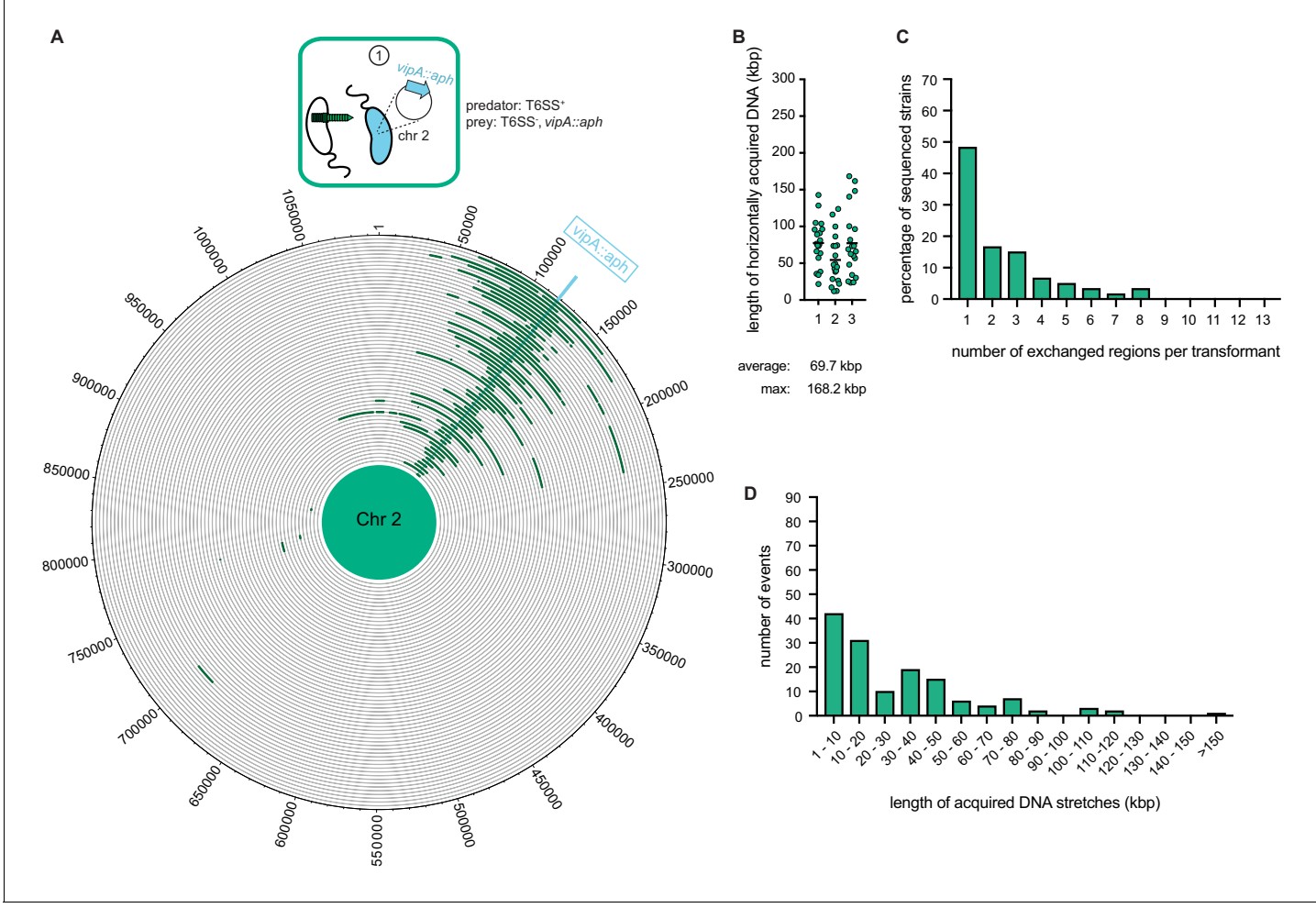

**Figure 3.** Whole-genome sequencing (WGS)-based quantification of horizontally acquired DNA. WGS analysis of transformants after prey killing and DNA transfer. Twenty kanamycin-resistant transformants were selected per independent biological experiment (n = 3). (**A**) The scheme represents the experimental setup of the co-culture experiment (condition ①). Sequencing reads for each transformant were mapped onto the prey genomes to visualize the transferred DNA regions (in dark green; see **Figure 4—figure supplement 2** for both chromosomes). The position of the resistance cassette (*aph*) is indicated by the light blue line. (**B**) Total DNA acquisition frequently exceeds 100 kb. The total length of horizontally acquired DNA is indicated on the Y-axis for each transformant. Data are from three biologically independent experiments as indicated on the X-axis. Average and maximum lengths are indicated below the graph. (**C**) Multiple transferred DNA regions were identified in the transformants. Percentage of transformants (n = 60) that exchanged one or more DNA regions, as indicated on the X-axis. (**D**) Large DNA stretches are transferrable by transformation. The length of individual consecutive DNA stretches was determined as indicated on the X-axis.

DOI: https://doi.org/10.7554/eLife.48212.008

The following source data is available for figure 3:

**Source data 1.** Raw data for **Figure 3**.

DOI: https://doi.org/10.7554/eLife.48212.009

mimicked by growing *S. pneumoniae* on synthetic biofilm-like assemblages on agar plates or on human lung epithelial cell lines. These conditions were used to recapitulate the natural nasopharynx environment in which this competent bacterium is frequently encountered (*Cowley et al., 2018*). Their data combined with our findings therefore contradict the notion that natural transformation is unlikely to serve DNA repair purposes or for the acquisition of new genetic information due to the quick fragmentation of free DNA in natural environments (*Nielsen et al., 2007*; *Overballe-Petersen et al., 2013*).

## Transformation by purified DNA only occurs if correctly timed

To better understand the DNA acquisition and integration potential of naturally competent *V. cholerae*, we next compared the data described above, which we refer to from now on as condition ① using experiments varying the aspects of neighbor predation and DNA supplementation (*Figure 4A*). First, the acceptor strain was grown in a monoculture immediately supplemented with purified genomic DNA (gDNA) derived from the same donor (prey) strain as described above. Notably, when the gDNA was added at the start of the chitin-dependent culture, no transformants were reproducibly detected from three independent biological experiments, suggesting that free DNA is rapidly degraded under such conditions. This finding is consistent with our previous work in which we demonstrated that *V. cholerae* produces an extracellular and periplasmic nuclease Dns (*Blokesch and Schoolnik, 2008*; *Seitz and Blokesch, 2014*) that degrades transforming material. At high cell density (HCD), where competence is induced, *dns* is partially repressed through direct binding of HapR (*Blokesch and Schoolnik, 2008*; *Lo Scrudato and Blokesch, 2013*), and this repression is reinforced by the transcription factor QstR (*Lo Scrudato and Blokesch, 2013*; *Jaskólska et al., 2018*). We therefore concluded that the simultaneous expression of both machineries, concomitantly with a strong repression of *dns*, is a prerequisite for successful DNA transfer. Indeed, such coordinated expression would ensure that T6SS-mediated attacks are exquisitely timed with low nuclease activity so that the prey-released DNA can be efficiently absorbed.

As we previously showed that the addition of purified gDNA after ~20–24 hr of growth on chitin wasn't prone to degradation by Dns (*Marvig and Blokesch, 2010*), we next choose this time point to probe the DNA acquisition capability using purified DNA (condition ②; *Figure 4A*). Doing so led to similar transformation frequencies as those observed for the prey-released DNA caused by T6SS attacks (condition ①; *Figure 4—figure supplement 1A*). WGS of 20 transformants from two biologically independent experiments likewise resulted in similar DNA acquisition patterns with average and maximum DNA acquisitions of 70 kbp and 188 kbp, respectively, and the presence of multiple exchanged regions of varying sizes (*Figure 4* and *Figure 4—figure supplements 2* and *3*). While we cannot entirely exclude that the maximum length of individual DNA stretches was biased by the purification step, despite the fact that we chose a method that was designed for chromosomal DNA isolation of 20–150 kbp sized fragments (see Materials and methods), our results suggest that the maximum DNA acquisition length of single fragments is probably reached between 100–110 kbp (*Figure 4—figure supplement 3*). Moreover, the comparable acquisition patterns between conditions ① and ② (*Figure 4*) imply that the prey-released DNA in condition ① is neither heavily fragmented nor is its accessibility or absorption by the competent acceptor bacterium significantly hindered due to, for example, DNA-binding proteins.

## Prey-exerted T6SS counter attacks do not change the DNA transfer pattern

Since the *aph* cassette was located within the T6SS sheath protein gene *vipA* in the above experiments, we wondered if this T6SS inactivation biased the DNA transfer efficiency. We therefore repeated the above-described experiments using prey strains that carried the *aph* cassette on the opposite site of chr 2 (within gene VCA0747; condition ③). As shown in (*Figure 4—figure supplement 1*), similar transformation frequencies were observed independent of the position of the *aph* cassette. Moreover, WGS of 2 × 20 transformants showed similar average and maximum DNA acquisition values (55.7 kbp and 227.4 kbp, respectively; *Figure 4*) as well as similar distribution patterns around the resistance marker (*Figure 4—figure supplement 4*). However, while not statistically supported, it appeared as if these conditions were prone to the acquisition of multiple non-connected regions, as transformants with only single/connected exchanges dropped from ~50% (*Figure 4—figure supplement 2* for condition ①) to around 20% (*Figure 4—figure supplement 4* for condition ③). Based on this observation, we hypothesized that the now-restored T6SS-mediated killing capacity of the prey led to the additional release of genomic DNA from the predator, which interfered with the uptake of prey-released DNA. To test this idea, we repeated condition ③ (e.g., *aph* within VCA0747) though again inactivated the T6SS of the prey using a non-selected marker (*cat*; condition ④), expecting the results to be similar to those of condition ① if this hypothesis was correct. No statistically significant differences were observed between both conditions (③ and ④) for all tested characteristics including transformation frequency (*Figure 4—figure supplement 1B*), number of

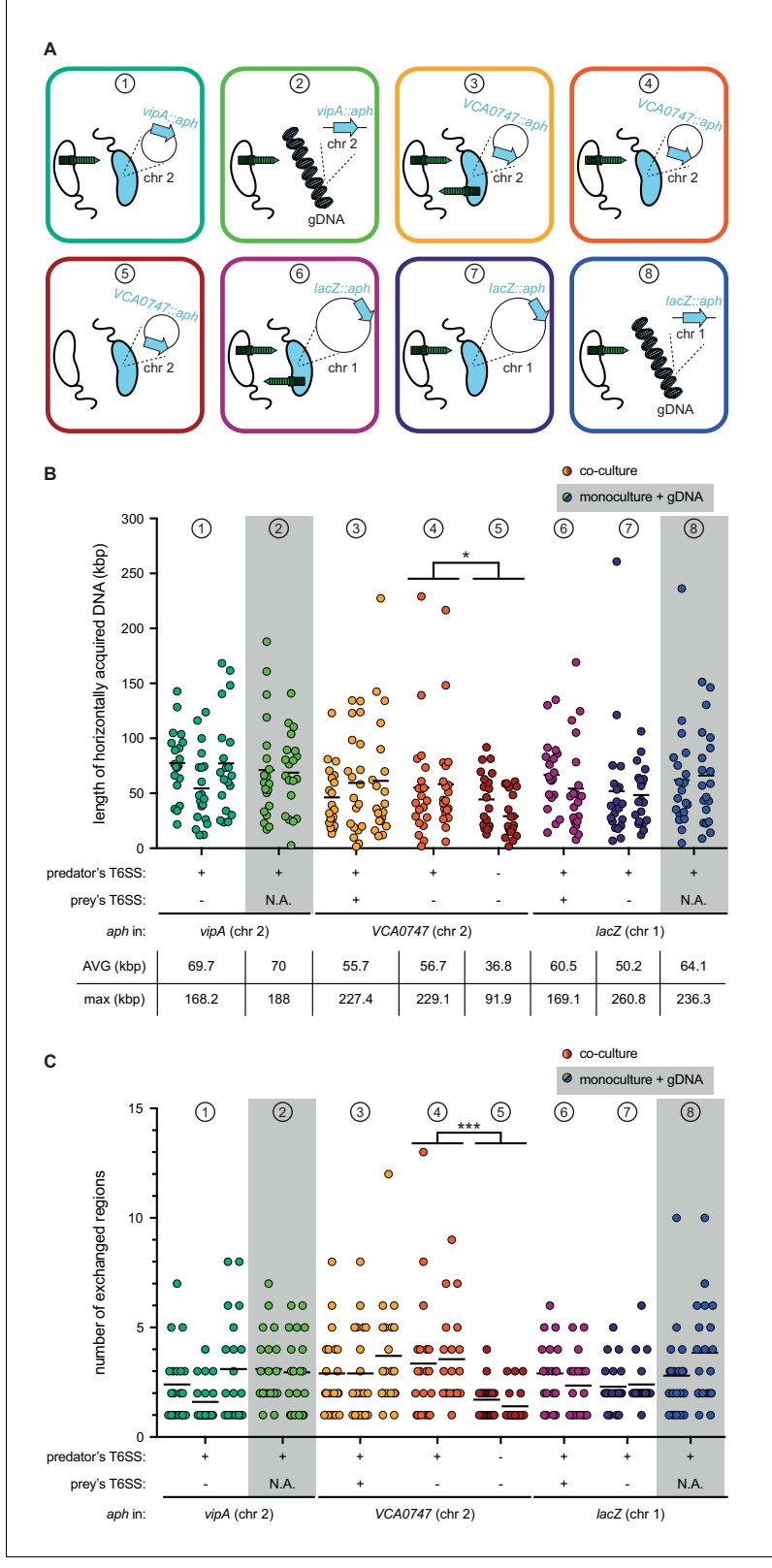

**Figure 4.** T6SS-mediated neighbor predation followed by DNA uptake enhances the frequency and length of transferred DNA stretches. (A) Scheme representing the eight experimental conditions tested in this study. Each scheme indicates whether the transformants acquired the *aph* resistance gene from a prey bacterium (blue) (position of *aph* indicated on the zoomed-in circles of chr 1 or chr 2) or from purified genomic DNA (gDNA). In the *Figure 4 continued on next page*

*Figure 4 continued*

former case, the killing capacity of the predator (white) and prey (blue) is shown by the presence or absence of the dark green T6SS structure. The same color code is maintained throughout all figures. (B–C) Transformants from independent biological experiments (n ≥ 2) were analyzed by WGS for each of the conditions ① to ⑧, as indicated at the top of each graph. The main features of predator and prey/gDNA are summarized below the X-axis. Panels (B) and (C) depict the total length of acquired DNA and the number of exchanged DNA stretches, respectively, for each transformant. N.A., not applicable. Statistical analysis is based on a pairwise comparison between different conditions. *p<0.05, ***p<0.001.
DOI: https://doi.org/10.7554/eLife.48212.010

The following source data and figure supplements are available for figure 4:

**Source data 1.** Raw data for *Figure 4*.
DOI: https://doi.org/10.7554/eLife.48212.021
**Figure supplement 1.** Natural transformation is enhanced by T6SS-mediated killing of prey bacteria.
DOI: https://doi.org/10.7554/eLife.48212.011
**Figure supplement 1—source data 1.** Raw data for *Figure 4—figure supplement 1*.
DOI: https://doi.org/10.7554/eLife.48212.012
**Figure supplement 2.** WGS-based quantification of horizontally acquired DNA under condition ①.
DOI: https://doi.org/10.7554/eLife.48212.013
**Figure supplement 3.** WGS-based quantification of horizontally acquired DNA under condition ②.
DOI: https://doi.org/10.7554/eLife.48212.014
**Figure supplement 4.** WGS-based quantification of horizontally acquired DNA under condition ③.
DOI: https://doi.org/10.7554/eLife.48212.015
**Figure supplement 5.** WGS-based quantification of horizontally acquired DNA under condition ④.
DOI: https://doi.org/10.7554/eLife.48212.016
**Figure supplement 6.** WGS-based quantification of horizontally acquired DNA under condition ⑤.
DOI: https://doi.org/10.7554/eLife.48212.017
**Figure supplement 7.** WGS-based quantification of horizontally acquired DNA under condition ⑥.
DOI: https://doi.org/10.7554/eLife.48212.018
**Figure supplement 8.** WGS-based quantification of horizontally acquired DNA under condition ⑦.
DOI: https://doi.org/10.7554/eLife.48212.019
**Figure supplement 9.** WGS-based quantification of horizontally acquired DNA under condition ⑧.
DOI: https://doi.org/10.7554/eLife.48212.020

exchanges, and separate and collective length (*Figure 4* and *Figure 4—figure supplements 4* and *5*), suggesting that predator-released DNA does not interfere with the predator's overall transformability by the prey-released DNA. However, we acknowledge that the technical limitations of the experimental setup did not allow the identification of complete revertants that first acquired and then again lost the *aph* cassette.

## T6SS-independent prey lysis rarely triggers DNA transfer and results in shorter DNA exchanges

We next tested whether T6SS-mediated DNA release impacted the length of the exchanged region, which would support the above speculation that the intimate co-regulation of type VI secretion, nuclease repression, and DNA uptake ensures that freshly released DNA is rapidly absorbed by the predator and is therefore less prone to fragmentation. Such co-regulation would not hold true for T6SS-independent DNA release as a result of random cell lysis, so we tested the transfer efficiency of the *aph* cassette under conditions in which both donor and acceptor strains were T6SS-defective (condition ⑤; *Figure 4* and *Figure 4—figure supplement 6*). Under such conditions, the transformation frequency dropped by 99.7% (*Figure 4—figure supplement 1B*), and WGS of 2 × 20 of these rare transformants showed significant differences. Indeed, the average and maximal length of acquired DNA (*Figure 4B*) and the number of exchanged regions (*Figure 4C*) were significantly different when T6SS+ versus T6SS- acceptor strains were compared, with the latter exchanges never exceeding four events compared to up to 13 events for T6SS-mediated DNA release (*Figure 4—figure supplements 5* and *6*). Based on these data, we conclude that T6SS-mediated DNA acquisition not only increases the transfer efficiency by ~100 fold but also fosters the exchange of multiple DNA stretches of extended lengths.

## T6SS-mediated DNA exchanges are not limited to the small chromosome

The experiments described above were designed to primarily score the transfer efficiency of DNA fragments localized on chr 2. The rationale behind this approach was a recent population genomic study on *Vibrio cyclitrophicus* that suggested the mobilization of the entire chromosome 2 and caused the authors to speculate: 'how often and by what mechanism are entire chromosomes mobilized?' (*Shapiro et al., 2012*). In the current study, we were unable to experimentally show such large transfer events. We considered four potential reasons for the absence of such large transfers: 1) mild fragmentation of prey-released DNA that excluded fragments above ~200 kb; 2) limited DNA uptake and periplasmic storage capacity of the acceptor strain (*Seitz and Blokesch, 2014*; *Seitz et al., 2014*); 3) limited protection of the incoming single-stranded DNA by dedicated proteins (such as Ssb and DprA; *Mortier-Barrière et al., 2007*; *Suckow et al., 2011*); or 4) lethality of larger exchanges due to the presence of multiple toxin/antitoxin modules within the integron island on chr 2 of *V. cholerae* (*Iqbal et al., 2015*). While technical limitations did not allow us to address the first three points, we followed up on the last idea by repeating the above-described experiments using prey strains in which the *aph* cassette was integrated on the large chromosome 1 (inside *lacZ*). We used these to test three (co-)culture conditions in which the prey strain was either T6SS-positive (condition ⑥), T6SS-negative (condition ⑦), or replaced by purified gDNA (condition ⑧; *Figure 4A*). As shown in *Figure 4—figure supplement 1*, the *aph* cassette was again transferred with high efficiency from the killed prey strain to the acceptor strain. However, comparing conditions ⑥ (co-culture conditions) and ⑧ (prey-derived purified gDNA as transforming material) revealed a small but significant transformation increase for the latter condition (~4 fold; *Figure 4—figure supplement 1A*). Based on these data, we speculate that the larger size of chr 1 (~3 Mb) compared to chr 2 (~1 Mb) slightly lowers the probability of acquiring the *aph* cassette when released from the killed prey that carried the resistance marker on chr 1. Indeed, the cassette's relative occupancy per chromosome is reduced by one third for the larger chromosome 1 compared to the smaller chromosome 2, which would reduce the chance of absorbing the *aph* cassette if it was embedded in large chromosomal fragment(s). This effect would become negligible, however, when purified gDNA is provided, most likely due to the size constraints of the purification procedure (max. 150 kb), which ultimately results in more or less equally sized DNA fragments. Consistent with this idea was the finding that purified gDNA from all those prey strains described in this study resulted in the same level of transformation no matter where the selective marker was located (*Figure 4—figure supplement 1D*).

Next, we randomly picked 20 transformants from two biologically independent experiments for each of these three experimental conditions (⑥ to ⑧) and sequenced their genomes (*Figure 4—figure supplements 7 to 9*). The analysis of these transformants showed that the average and maximum DNA acquisition values were highly comparable to those described above for DNA exchanges on chromosome 2 (*Figure 4B*) and that multiple exchanged regions were likewise observed (*Figure 4C*). We therefore conclude that prey-derived transforming DNA can equally modify both chromosomes. Moreover, our data suggest that consecutive stretches of exchanged DNA above ~200 kbp either do not occur or occur at levels below the detection limit of this study, and that this size limitation is not caused by the toxin/antitoxin–module-containing integron island on chr 2.

## Conclusion

Based on the data presented above, we conclude that T6SS-mediated predation followed by DNA uptake leads to the exchange of large DNA regions that can bring about bacterial evolution. This finding is consistent with the heterogeneous environmental *V. cholerae* populations that were observed in cholera-endemic areas (*Faruque et al., 2004*). Still, an open question that remains is why pandemic cholera isolates are seemingly clonal in nature (*Mutreja et al., 2011*; *Domman et al., 2017*; *Weill et al., 2017*; *Weill et al., 2019*), and we propose two explanations for this. First, sampling strategies might be biased for the selection of the most pathogenic strains and, concomitantly, exclude less virulent variants that have undergone HGT events. Secondly, transformation-inhibiting nucleases similar to Dns (*Blokesch and Schoolnik, 2008*) have recently spread throughout pandemic *V. cholerae* isolates as part of mobile genetic elements (experimentally shown for VchInd5 [*Dalia et al., 2015*] and predicted for SXT [*Blokesch, 2017*]), which makes these pandemic strains

less likely to undergo HGT events. One could also argue that pandemic *V. cholerae* are rarely exposed to competence-inducing chitinous surfaces due to the prevalence of inter-household transmission throughout cholera outbreaks (*Clemens et al., 2017*). Yet in vivo-induced antigen technology (IVIAT) assays showed strong human immune responses against proteins of the DNA-uptake pilus that fosters natural transformation, kin recognition, and chitin colonization (*Meibom et al., 2005*; *Seitz and Blokesch, 2013*; *Adams et al., 2019*), which contradicts this idea. Indeed, the major pilin of the DNA-uptake pilus, PilA, was most frequently identified by IVIAT together with the outer-membrane secretin PilQ (*Hang et al., 2003*), which suggests that the bacteria encounter competence-inducing conditions either before entering the human host or after its colonization. The latter option is not, however, supported by in vivo expression data from human volunteers (*Lombardo et al., 2007*). Notably, our work shows the incredible DNA exchange potential that chitin-induced *V. cholerae* strains exert under co-culture conditions and future studies are therefore required to better understand strain diversity in clinical and environmental settings in the absence of sampling biases.

# Materials and methods

## Key resources table

| Reagent type (species) or resource | Designation | Source or reference | Identifiers | Additional information |
|---|---|---|---|---|
| Strain, strain background (*Vibrio cholerae*) | *V. cholerae* O1 El Tor, strain A1552 (Primary strain) | PMID: 9473029 PMID: 30574591 (genome sequence) | | See *Supplementary file 1* for strains and plasmids used in this study |
| Strain, strain background (*Vibrio cholerae*) | *V. cholerae*, strain Sa5Y (Secondary strain) | PMID: 17449702 PMID: 30574591 (genome sequence) | | See *Supplementary file 1* for strains and plasmids used in this study |
| Other | TCBS agar, selective medium | Sigma-Aldrich | 86348–500G | Additional, standard growth media are described under growth conditions |
| Peptide, recombinant protein | Pwo SuperYield DNA Polymerase | Roche / Sigma-Aldrich | 4340850001 | |
| Peptide, recombinant protein | GoTaq G2 DNA Polymerase | Promega | M7848 | |
| Commercial assay or kit | Genomic-tip 100/G (DNA purification) | Qiagen | 10243 | |
| Commercial assay or kit | Genomic DNA Buffer Set | Qiagen | 19060 | |

## Bacterial strains, plasmids, and growth conditions

The bacterial strains and plasmids used in this study are described in *Supplementary file 1*. Bacteria were routinely grown aerobically in lysogeny broth (LB) or on LB agar plates (1.5% agar) at 30°C or 37°C. Half-concentrated defined artificial seawater medium (0.5x DASW) containing HEPES and vitamins (*Meibom et al., 2005*) was used for growth on chitinous surfaces for chitin-induced T6SS killing and natural transformation experiments (as previously described; *Marvig and Blokesch, 2010*; *Borgeaud et al., 2015*). Agar plates containing M9 minimal medium (Sigma-Aldrich) supplemented with vitamins (MEM vitamin solution; Gibco), 0.001% casamino acids (Merck), and 0.2% mannose were used to select *V. cholerae* strain A1552 and to exclude strain Sa5Y (to check the direction of transformation for *comEC*-positive prey). Thiosulfate-citrate-bile salts-sucrose (TCBS; Sigma-Aldrich) agar plates were used to counterselect *E. coli* strains after mating with *V. cholerae*. Antibiotics were used at the following concentrations whenever required: chloramphenicol (Cm), 2.5 µg/ml; kanamycin (Kan), 75 µg/ml; streptomycin (Strep), 100 µg/ml; ampicillin (Amp), 100 µg/ml; and rifampicin (Rif), 100 µg/ml.

## DNA manipulation techniques

Recombinant DNA techniques were performed following standard molecular–biology-based protocols (*Sambrook et al., 1982*). DNA-modifying enzymes such as Pwo DNA polymerase (Roche), Taq DNA polymerase (GoTaq; Promega), and restriction modification enzymes (New England Biolabs) were used according to the manufacturer's recommendations. Genetically modified strains were verified by colony PCR and, if required, also by Sanger sequencing (Microsynth, Switzerland) for their correctness.

## Genetic engineering of bacterial strains

To delete gene(s) from the parental WT strains (A1552 or Sa5Y), a gene-disruption method based on either a counter-selectable suicide plasmid pGP704-Sac28 (*Meibom et al., 2004*) or on natural transformation and FLP recombination was used (TransFLP method; *De Souza Silva and Blokesch, 2010*; *Blokesch, 2012b*; *Borgeaud and Blokesch, 2013*). Natural transformation was also used to insert the antibiotic resistance cassettes *aph* (KanR), *cat* (CmR), and/or *bla* (AmpR) into target gene(s) of *V. cholerae*.

## Preparation of genomic DNA

Genomic DNA (gDNA) was purified from a 2 ml culture of the respective strain. DNA extraction was performed using 100/G Genomic-tips together with a Genomic DNA buffer set as described in the manufacturer's instructions (Qiagen). After precipitation, the DNA samples were transferred into Tris buffer (10 mM Tris-HCl, pH 8.0). This was preferred over rapid gDNA isolation kits such as the DNeasy Blood and Tissue kit (Qiagen), as the latter isolation kit is strongly biased towards shorter DNA fragments (predominantly 30 kb in length compared to up to 150 kb for the 100/G columns, as stated by the manufacturer).

## Natural transformation assay

Natural transformation assays were performed by adding purified gDNA to the chitin-grown bacteria or by co-culturing the two non-clonal *V. cholerae* strains. To set up the experiments, the bacterial strains were grown as an overnight culture in LB medium at 30°C. After back dilution, the cells were incubated in the presence of chitin flakes (~80 mg; Sigma-Aldrich) submerged in a final volume of 1 ml of half-concentrated (0.5x) defined artificial seawater medium (*Meibom et al., 2005*). When purified DNA served as the transforming material, 2 µg of the indicated gDNA (final concentration 2 µg/ml) was added after 24 hr of growth on chitin (except for the experiments in which the gDNA was added at 0 hr, as indicated in the text), and the cells were incubated for another 6 hr. At that point, the bacteria were detached from the chitin surfaces by vigorous vortexing and then were serially diluted. Colony-forming units (CFUs) were enumerated on selective (antibiotic-containing) or non-selective (plain LB) agar plates, and the transformation frequency was calculated by dividing the number of transformants by the total number of CFUs.

For mixed community assays, the two strains were inoculated simultaneously at a ratio of 1:1 in a final volume of 1 ml. We estimated that if all of the inoculated prey/donor cells would lyse,~0.2 µg/ml of DNA would be released into the medium. These mixtures were incubated for 30 hr before the bacteria were harvested, diluted, and plated, as described above (maximum prey/donor DNA that could be released at this point corresponds to ~0.4 µg/ml).

All transformation frequency values are averages of three biologically independent experiments except for WGS conditions ② and ④–⑧, wherein the averages of two independent experiments are depicted.

## Genome comparisons

Each chromosome was segmented in contiguous fragments of 10 kb, which were locally aligned against the corresponding chromosome of a reference genome. Each fragment was aligned in the forward and reverse orientation, and the best alignment was retained. The number of differences per 10 kb was evaluated by counting the number of events necessary to mutate the reference to obtain the 10 kb fragment (e.g. an insertion or deletion of an arbitrary number of nucleotides would count as one event). To visualize the overall architecture and differences, circular plots in which each 10 kb fragment was linked to its reference genome counterpart and colored according to the

number of differences were made in R using the circlize package (*Gu et al., 2014*). Black and blue linkers indicated whether the 10 kb fragment had the same or reverse orientation relative to the reference.

As a recent study identified a large inversion close to the origin of replication of chromosome 1 in the original genome sequence of N16961 (*Val et al., 2016*), which most likely resulted from an imperfect assembly or a lab domestication event, we sequenced and de novo assembled the genome of our laboratory stock of strain N16961 (*Matthey et al., 2018*). This stock is resistant to streptomycin, which is consistent with most literature reports on strain N16961, while the original reference strain N16961 remained sensitive to streptomycin according to its genome sequence. This difference suggests that mutagenesis event(s), including the characteristic streptomycin-resistance mutation within *rpsL* (encoding for RpsL[K43R]) must have occurred while the strain was domesticated. Comparative genomics between our laboratory stock of N16961 and the reference genome (*Heidelberg et al., 2000*) showed a high level of sequence identity, though also confirmed the previously reported inversion around the origin of chromosome 1 (*Val et al., 2016*) (*Figure 2—figure supplement 1*). Additionally, significant differences were observed between the two N16961 genome sequences with respect to the number and arrangement of the ribosomal RNA clusters, which could have resulted from assembly artifacts. Indeed, while Heidelberg and colleagues described the presence of eight rRNA operons (16S-23S-5S) (*Heidelberg et al., 2000*), we found ten rRNA operons (plus an additional 5S copy close to the tRNA-Thr). The only other major differences observed between the strains were in genes VC1620 and *vasX* (VCA0020) on chromosome 1 and 2, respectively, as also mentioned in the main text (*Figure 2—figure supplement 1*; marked with * and #). VC1620, or *frhA,* on chr 1 encodes a large protein with several cadherin tandem repeat domains, the number of which vary among different *V. cholerae* strains (*Syed et al., 2009*). Due to the repetitive nature of this DNA region, an assembly mistake in either of the two genome assemblies can therefore not be excluded. The sequence of the newly sequenced stock of N16961 was, however, identical to the region in pandemic strain A1552 (*Figure 2—figure supplement 2*). For the discrepancy within *vasX* on chr 2, we observed a significant number of single nucleotide polymorphisms (SNPs; n = 11) or nucleotide insertions (n = 14) within this 3,272-bp-long gene. One of these nucleotide insertions caused a frameshift that therefore resulted in a premature stop codon. As BLAST analyses suggested that this change was strain specific and not previously observed, we Sanger sequenced the corresponding DNA region using the same genomic DNA preparation that was used for the long-read PacBio sequencing, which allowed us to elucidate whether the mutations reflected a lab domestication event or whether the PacBio sequence was of low quality in this specific area of the genome. The latter turned out to be the case, as the mutations described above were absent from the Sanger sequencing reads. We therefore conclude that the two genomes are highly identical and that the major differences involve imperfect rRNA cluster assembly in the reference genome, which resulted in the underestimation of the number of rRNA clusters and the inverted assembly around the origin of replication of chr 1.

*V. cholerae* O1 El Tor (Inaba) strain A1552 (formerly known as 92A1552-Rif^r) is a rifampicin-resistant derivative of strain 92A1552 (*Yildiz and Schoolnik, 1998*), which was isolated in California from a traveler returning from South America. Epidemiological investigations concluded that the transmission of this strain occurred via a contaminated seafood salad that was served on an airplane between Lima, Peru and Los Angeles, California (*Eberhart-Phillips et al., 1996*; *Blokesch, 2012a*), which links this strain to the Peruvian cholera outbreak in the 1990s.

Comparing the de novo assembled genome sequence of A1552 (*Matthey et al., 2018*) to the genome of our laboratory stock of N16961 showed a large degree of genomic conservation between both isolates despite the fact that strain A1552 contained only nine rRNA operons (16S-23S-5S) instead of the ten operons in strain N16961. The genome architecture, however, was not maintained, as strain A1552 had undergone a large inversion of approximately 2.7 Mbp between two rRNA clusters on chr 1 (*Figure 2—figure supplement 2*; genome sequence inverted to simplify visualization), a finding that is consistent with a previous report (*Kemter et al., 2018*). Apart from this inversion, the major differences between both strains were the presence of the WASA-1 island (a 44 genes-carrying element of unknown function; *Mutreja et al., 2011*) and a modified *Vibrio* seventh pandemic island II (VSP-II) in strain A1552 (*Figure 2—figure supplement 2*). These features allowed us to classify strain A1552 as belonging to the LAT-1 sublineage of the West-African South

American (WASA) lineage of the seventh pandemic *V. cholerae* strains (*Mutreja et al., 2011*; *Domman et al., 2017*).

## Whole-genome sequencing of transformants

For WGS, transformation assays were performed as described above using eight different experimental conditions (*Figure 4* and listed in *Supplementary file 2*). To focus on the acquisition potential of strain A1552, conditions ⑤–⑧ used transformation-deficient prey strains (e.g., Sa5Y derivatives in which a *bla* cassette interrupted the DNA translocation channel protein encoding gene *comEC*; *Seitz and Blokesch, 2013*). The 360 recovered transformants (3 × 20 for experimental conditions ① and ③, which showed high levels of reproducibly, followed by 2 × 20 for all other conditions; see *Supplementary file 2*) were grown overnight in LB medium. Genomic DNA extraction was performed as described above. Further processing of the samples was conducted by Microsynth (Balgach, Switzerland). The quality of the DNA samples was verified before DNA libraries were prepared using a Nextera XT Library Prep kit (Illumina). Paired-end sequencing was performed using a NextSeq 500 sequencer (Illumina) with read lengths of 75 nt resulting in mean fragment lengths of around 200 nucleotides.

## Scoring of horizontal gene transfer events through bioinformatics analyses

HGT events were scored for the 360 transformants that were derived from the eight different experimental conditions (*Figure 4* and listed in *Supplementary file 2*). For each condition, a predator/ acceptor strain (A1552 or its derivative) and a prey/donor strain (derivatives of Sa5Y; used in mixed cultures or as purified gDNA) were defined, and their genomic sequences were generated in silico. These in silico templates were based on the recently announced genome sequences of the parental strains (*Matthey et al., 2018*) to which the integrated genomic features were added (e.g., integration of *aph*, *cat*, and/or *bla* cassettes proceeded by constitutive promoters). Each DNA template contained two parts reflecting the large ~3.0 Mbp chr 1 and the small ~1.1 Mbp chr 2. Preliminary analyses identified the presence of systematic differences in each sample, which can be attributed to errors in the reference templates. Thus, before starting the final analysis process, the following patches were applied to the chr 2 of the predator/acceptor reference genomes:

- Coordinate (in reference genome CP028895): 445515–445520 => TTTTTT replaced by TTTTT.
- Coordinate (in reference genome CP028895): 447288–447289 => GC replaced by G.
- Coordinate (in reference genome CP028895): 467939–467940 => AT replaced by A (resulting in a replacement of TTTTTTT [467940–467946] by TTTTTT).

The correctness of these in silico changes was confirmed by Sanger sequencing.

The FASTA sequences of the corrected reference templates used for this work are available in the Reference directory on GitHub: https://github.com/sib-swiss/VibrioCholerae_HGT (*Iseli, 2019*; copy archived at https://github.com/elifesciences-publications/VibrioCholerae_HGT).

Each read pair was aligned against both genomes of the donor and acceptor strains, and the position with the least number of mismatches was kept. If multiple possibilities with the same number of mismatches were possible, the possibilities were kept for later processing. The alignments were performed with an in-house code derived from the fetchGWI tool [https://sourceforge.net/projects/tagger/files/fetchGWI-tagger/], but other tools such as BWA or bowtie2 would equally qualify for the same purpose. The results of this first analysis phase were obtained as tab-delimited lists of read pairs with their alignment positions, number of mismatches for each of the reads of the pair, and number of matching positions with the same number of mismatches.

The second step of the analysis consisted of a C program that parsed the tab-delimited file of the previous step and recorded the accumulated coverage and the observed nucleotide for each position of each of the donors' and acceptors' chromosomes. For each analysis, two separate recordings were kept: the first recording took only those read pairs into account that showed zero mismatches together with a unique unambiguous matching position, while the second recording took all other matches into account (e.g., a coverage of 1 was counted in each matched position regardless of whether the match was unique or not). At the end of this process, regions with continuous coverage were created by looking for positions in the first recording (unique and exact matches) for which the observed coverage was at least 3. The start and end positions of such continuous stretches were

determined by extending as far as possible from this seed position, taking into account the total coverage from both recordings. If the length of the so-defined fragment was at least 500 nucleotides it was kept for output. The output consisted of a FASTA file containing all the defined fragments and the FASTA header of each fragment. This header recorded which reference was covered (e.g., which strain [acceptor or donor] and which chromosome [large chr 1 or small chr 2]). For each position, the nucleotide that composed more than half of the coverage of that position was generated in the output. If no single nucleotide represented more than half of the coverage, an N was used.

The expectation was that the fragments obtained in the previous step constituted a full coverage of the transformant, which had inherited its genomic material mostly from the parental acceptor strain with some DNA regions originating from the donor strain by HGT; this referred particularly to those regions containing the antibiotic resistance cassette, which was used as selective marker. A global DNA alignment for each output fragment was therefore performed by mapping the fragments onto the corresponding chromosomes of the donor and acceptor strains. Since we determined that the donor and acceptor genomes differed on average by 1 nucleotide every 55 nucleotides, it was expected that several mismatches would be found when a fragment originating from the donor strain was aligned to the genome of the acceptor strain. The first and last mismatches of those fragments were defined as outer bounds of the transferred fragments, as they arguably represented the minimal length of the transferred DNA region. The summary data of these alignments and boundary information were determined using Perl scripts as were those regions of the acceptor genomes that were not covered by reads derived from the analyzed transformant (e.g., transformed acceptor strain). The output of those scripts (available on GitHub: https://github.com/sib-swiss/VibrioCholerae_HGT) was then used to generate a final table of transferred segments as well as summary plots by applying the R package circlize (*Gu et al., 2014*). Finally, the data were transferred to Excel to calculate the total length of horizontally acquired DNA per transformant as well as the number of HGT events per transformant. The GraphPad Prism software was used for graphic visualization. To validate the bioinformatic approach, the mapped reads obtained for >50 samples were also visually inspected for transferred regions using the software Geneious.

## Statistics

Statistically significant differences were determined by the two-tailed Student's *t*-test where indicated. For natural transformation assays, data were log-transformed (*Keene, 1995*) before statistical testing. When the number of transformants was below the detection limit, the value was set to the detection limit to allow for statistical analysis.

## Data availability

WGS reads of the 360 transformants have been deposited in NCBI's Sequence Read Archive (SRA) under SRA accession numbers SRR6934824 to SRR6935183 according to *Supplementary file 3*. The Bioproject accession number is PRJNA447902.

## Acknowledgements

The authors thank members of the Blokesch laboratory and F Le Roux for discussions and A Boehm for strain Sa5Y. We also acknowledge preliminary bioinformatic analyses by S Strempel (Microsynth), A-C Portmann, and I Mateus, who also uploaded the sequencing reads to NCBI. This work was supported by EPFL intramural funding, the Swiss National Science Foundation grant 31003A_162551, and a Starting (309064-VIR4ENV) and Consolidator (724630-CholeraIndex) grant from the European Research Council to MB. MB is a Howard Hughes Medical Institute (HHMI) International Research Scholar (grant #55008726).

## Additional information

### Funding

| Funder | Grant reference number | Author |
|---|---|---|
| Swiss National Science Foundation | 31003A_162551 | Melanie Blokesch |

| Seventh Framework Programme | 309064-VIR4ENV | Melanie Blokesch |
| H2020 European Research Council | 724630-CholeraIndex | Melanie Blokesch |
| Howard Hughes Medical Institute | 55008726 | Melanie Blokesch |

The funders had no role in study design, data collection and interpretation, or the decision to submit the work for publication.

## Author contributions

Noémie Matthey, Conceptualization, Formal analysis, Validation, Investigation, Methodology, Writing—review and editing; Sandrine Stutzmann, Candice Stoudmann, Investigation, Methodology; Nicolas Guex, Christian Iseli, Software, Formal analysis, Validation, Investigation, Methodology, Writing—review and editing; Melanie Blokesch, Conceptualization, Resources, Formal analysis, Supervision, Funding acquisition, Validation, Investigation, Methodology, Writing—original draft, Project administration, Writing—review and editing

## Author ORCIDs

Noémie Matthey (iD) http://orcid.org/0000-0002-6056-2756
Christian Iseli (iD) https://orcid.org/0000-0002-2296-2863
Melanie Blokesch (iD) https://orcid.org/0000-0002-7024-1489

## Decision letter and Author response

Decision letter https://doi.org/10.7554/eLife.48212.029
Author response https://doi.org/10.7554/eLife.48212.030

# Additional files

## Supplementary files

• Supplementary file 1. Strains and plasmids used in this study.
DOI: https://doi.org/10.7554/eLife.48212.022

• Supplementary file 2. Details of eight experimental conditions and corresponding strain numbers.
DOI: https://doi.org/10.7554/eLife.48212.023

• Supplementary file 3. Sequence Read Archive (SRA) submission details.
DOI: https://doi.org/10.7554/eLife.48212.024

• Transparent reporting form
DOI: https://doi.org/10.7554/eLife.48212.025

## Data availability

Sequencing reads have been deposited in NCBI's Sequence Read Archive (SRA) under SRA accession numbers SRR6934824 to SRR6935183. The Bioproject accession number is PRJNA447902.

The following previously published dataset was used:

| Author(s) | Year | Dataset title | Dataset URL | Database and Identifier |
|---|---|---|---|---|
| Matthey N, Drebes Dörr NC, Blokesch M | 2018 | Long-Read-Based Genome Sequences of Pandemic and Environmental Vibrio cholerae Strains | https://www.ncbi.nlm.nih.gov/bioproject/PRJNA447902 | NCBI Bioproject, PRJNA447902 |

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
