## [Decision Letter]

Thank you for submitting your article "Neighbor predation linked to natural competence fosters the transfer of large genomic regions in *Vibrio cholerae*" for consideration by *eLife*. Your article has been reviewed by two peer reviewers, and the evaluation has been overseen by a Reviewing Editor and Detlef Weigel as the Senior Editor. The reviewers have opted to remain anonymous.

The reviewers have discussed the reviews with one another and the Reviewing Editor has drafted this decision to help you prepare a revised submission.

This manuscript is extending the observations published by your group in 2015, where you showed that the *V. cholerae* T6SS, which serves as a predatory killing device, is part of the competence regulon in this bacterium. You had also shown that deliberate killing of nonimmune cells via competence-mediated induction of T6SS released DNA and made it accessible for horizontal gene transfer, but the precise extent of HGT remained unknown, as did the size of the DNA that could be taken up and recombined in conditions such as the ones used in the current study, which are closer to what happens in the real world than the conditions used in other previous studies on different competent models.

These questions are now expertly clarified with data gathered from the different experiments presented here. The studies answer several pending questions that are of interest for a large community of microbiologists interested in the bacterial evolutionary mechanism of HGT and its broader impact. In particular, you established that the size of DNA that could be taken up under the investigated conditions is remarkable, even more than 150kb, which is much larger than what was considered so far. You also show that in many cases there are multiple DNA fragments from different genome locations that are integrated by homologous recombination in a single clone – you exactly mapped all of these events by establishing the genome sequence of multiple clones in a set of various conditions and controls.

There are nevertheless a number of points that need to clarified before the paper can be accepted:

1) It is interesting that the authors observe that two markers cannot be transferred when they are on separate chromosomes during T6SS-mediated HGT but can be transferred when purified gDNA is used. In particular, in the subsection “T6SS-mediated DNA exchanges are not limited to the small chromosome”, in the discussion on the effect of the size of chr1 and chr2 on the frequency of the *aph* transfer is not clear. Each cell only carries one copy of each chromosome, so there should be only one *aph* equivalent par genome, whatever is the hosting chromosome. However, the different figures show many co-transformants carrying chr1 fragments when *aph* is on chr2, and vice versa. Possible explanations should be discussed in more details in the manuscript.

2) The authors concluded that T6SS-independent prey lysis results in exchange of shorter DNA fragments. To support this idea, they calculated the average and maximal length of acquired DNA and the number of exchanged regions under conditions where both donor and acceptor strains were T6SS-. However, such condition resulted in much less transformants. Another control condition would be to mix a T6SS-negative acceptor with a donor strain that can be lysed by for example inducing a phage lytic gene carried on a plasmid to lyse the donor cells in T6SS-independent manner.

3) The authors quantitatively compared the transfer of DNA under co-culture conditions or monoculture with purified gDNA. An estimation of DNA concentration released upon T6SS-mediated killing in the co-culture should be provided. Similarly, instead of total amount of gDNA added, authors should estimate what was the concentration of gDNA in the monoculture experiments. Are these comparable?

4) Statistical analysis of differences between data provided in Figure 1B and 1C should be provided to support certain claims in the text.

---

## [Author Response]

[…] There are a number of points that need to clarified before the paper can be accepted:1) It is interesting that the authors observe that two markers cannot be transferred when they are on separate chromosomes during T6SS-mediated HGT but can be transferred when purified gDNA is used. In particular, in the subsection “T6SS-mediated DNA exchanges are not limited to the small chromosome”, in the discussion on the effect of the size of chr1 and chr2 on the frequency of the aph transfer is not clear. Each cell only carries one copy of each chromosome, so there should be only one aph equivalent par genome, whatever is the hosting chromosome. However, the different figures show many co-transformants carrying chr1 fragments when aph is on chr2, and vice versa. Possible explanations should be discussed in more details in the manuscript.

We are grateful for this comment and discussed these points in more details in the manuscript, as suggested. We would like to note though that the comment addressed two different aspects: First, that two markers are rarely transferred under co-culture conditions when they are located on different chromosomes (Figure 1) and second, that the transfer efficiency is slightly lower when the *aph* cassette is integrated into chr 1 compared to chr 2 (Figure 4—figure supplement 1).

Concerning the second point: It is of course true that each cell carries only one copy of each chromosome. However, their sizes differ with the large chromosome (chr 1) being ~3 Mbp and the small chromosome (chr 2) only 1 Mbp. As such, the *aph* cassette occupies a smaller proportion of the total chr 1 compared to the total chr 2 (a ~3-fold difference). Thus, if major parts of the chromosomes are released and more or less kept intact while the transfer occurs, the chances for the *aph* acquisition is most likely reduced to one third if the cassette is located on chr 1 compared to chr 2 due to this size difference. This could explain the slightly lower transformation frequency. However, the chromosomal size difference would become irrelevant when purified DNA is provided, as the purified DNA is equally fragmented into DNA stretches of max. 150 kb. However, please note that we didn’t want to imply in any way that this 4-fold difference is of major biological significance. Instead, this part of the Discussion primarily serves to address the statistical difference that we observed and to provide a potential explanation of the latter, especially when compared to the purified DNA data.

Concerning the first point (Figure 1): Co-transfers indeed occur – also in trans (that is, on the other chromosome) though not in the majority of transformants and at vastly different positions. The transfer of two specific loci (e.g., the two resistance markers), however, will again depend on the location of the two cassettes and whether the fragments in which they are integrated are saturating the system or not. This is unlikely to be the case for purified gDNA, due to the size limitation that resulted from the purification method.

Overall, it should be noted though that the experiments described in Figure 1 were meant to introduce the question and provide the first hint that we are dealing with the transfer of large DNA regions, an idea that is then further addressed throughout the rest of the manuscript. Indeed, as discussed in the text, these data do not directly show whether one large DNA fragment was absorbed and integrated or two independent events occurred with the exception of the condition in which both resistance markers are on separate chromosomes (which by definition requires independent transfers). This scenario and the data we obtained is one of the most intriguing parts of Figure 1, as double resistance rarely occurs in co-culture experiments in which T6SS-mediated DNA release is involved, which is in contrast to gDNA supplementation conditions. This difference indirectly suggests that the prey-released DNA is saturating the system and that the released DNA is probably not fragmented to the same extent as the purified DNA (e.g., below 150kb in size). To highlight this point even better and to also address point #4 mentioned below, we now included an additional supplementary figure in which we directly compare the transformation frequencies of Figure 1B and 1C. Please note that we did not want to replace the initial panels 1B and 1C by the combined figure, as we realized that readers could get slightly lost if confronted with all the different conditions in a single panel. We hope that this is acceptable for the reviewers and editors.

2) The authors concluded that T6SS-independent prey lysis results in exchange of shorter DNA fragments. To support this idea, they calculated the average and maximal length of acquired DNA and the number of exchanged regions under conditions where both donor and acceptor strains were T6SS-. However, such condition resulted in much less transformants. Another control condition would be to mix a T6SS-negative acceptor with a donor strain that can be lysed by for example inducing a phage lytic gene carried on a plasmid to lyse the donor cells in T6SS-independent manner.

We very much appreciate this comment. And while we agree that the proposed experiment would be exciting from a genetic engineering standpoint, the results would be i) strongly biased and ii) not provide any novel information about the natural system that we aimed at elucidating in this study. Please let us explain these two points further. Why biased? Because the timing of induction of the plasmid-encoded suicide toxin gene (such as a lytic phage-derived lysis gene) would strongly determine when the prey DNA is released. This might influence transformation in a comparable manner as we showed for the supplemented gDNA experiments. Indeed, the released DNA would be quickly degraded at early time points (e.g., when *V. cholerae*’s nuclease Dns is produced and active) while it might serve as transforming material at high cell densities. However, in the latter case, the system would be further biased, as the inducer might not reach all cells that are deeply embedded in chitin-attached biofilms equally compared to those that are surface-exposed or planktonic and therefore not in close contact with an acceptor strain. Hence, three different outcomes can be envisioned from the proposed experiment: first, the transformation data would look similar to T6SS-mediated prey lysis; second, the data would mimic the random lysis data, or, due to enhanced DNA release, the supplemented gDNA data; or third, the data would look completely different from the all other scenarios. We truly believe that none of these potential outcomes would contradict the current manuscript and neither strengthen the overall conclusions of our study, as such a genetically engineered prey suicide system is hugely artificial and not driven by the acceptor/predator strain itself. Indeed, the outcome would primarily dependent on the initial optimization steps apart from the fact that setting up such an artificial system would be very time consuming (estimated as around 6 months for the cloning, bringing it onto the chromosomes (as *V. cholerae* doesn’t like plasmids), induction optimization protocol without and ultimately with chitinous surfaces, the real experiment in three independent biological replicates, the isolation of the transformants, high quality DNA extractions from 40-60 transformants, NGS, and the data processing and analysis) and costly. We therefore disagree that this experiment would represent a “control condition” and respectfully refrain from doing it, as we truly believe that the results would not provide novel insight on the natural system of T6SS-mediated killing combined with DNA uptake.

The point that transformants are rare under such random lysis conditions is true and, in fact, of prime importance. Indeed, we strongly emphasized this point in the manuscript as the combination of seeing that transformants are rare (e.g., 1% compared to interbacterial killing conditions) and that these rare transformants carry less transferred material gives strong indication of the benefit that arises from coupling the two systems (T6SS-mediated killing and DNA uptake). Hence, it is transformation quantity and quality that changes, which is what we emphasized in the manuscript.

3) The authors quantitatively compared the transfer of DNA under co-culture conditions or monoculture with purified gDNA. An estimation of DNA concentration released upon T6SS-mediated killing in the co-culture should be provided. Similarly, instead of total amount of gDNA added, authors should estimate what was the concentration of gDNA in the monoculture experiments. Are these comparable?

This is again a very valid and interesting point. Indeed, the amount of gDNA that we added was based on optimized protocols that we established over the past 15 years. However, as suggested, we now included more information in the revised manuscript and added estimations of how much prey DNA might be maximally released. Briefly, the concentration of supplemented gDNA was 2 µg for all corresponding experiments. Due to a final volume of 1 ml (information added to manuscript) the final concentration was therefore 2µg/ml. For the coculturing conditions we now provided two estimates in the manuscript: the first one is based on the inoculum (e.g., if all prey cells that we initially added into the mixture would lyse). This value corresponds to ~0.2µg/ml. The second estimate is based on the number of *V. cholerae* cells that can be recovered after incubation on chitin for 30h (~0.4µg/ml). Thus, even if all prey cells would lyse, which is unlikely to occur as many of these cells will not be in direct contact with a predatory bacterium, the total amount of prey-released DNA would still be lower than the supplemented gDNA in the corresponding experiments. These data therefore further strengthen the findings that neighbor predation strongly enhances HGT.

4) Statistical analysis of differences between data provided in Figure 1B and 1C should be provided to support certain claims in the text.

We fully agree with this point and thank the reviewers for bringing it up. We now added a new supplementary figure that combines the panels 1B and 1C, as described above under point #1. Statistical analysis has been performed for these comparative data and the differences are indicated. As the reviewers can now appreciate, the statistical analysis fully supports our claims.